# The sterol C-24 methyltransferase encoding gene, *erg6*, is essential for viability of *Aspergillus* species

Jinhong Xie [1,2], Jeffrey M. Rybak [3], Adela Martin-Vicente[2], Xabier Guruceaga [2], Harrison I. Thorn [1,2], Ashley V. Nywening[2,4,5], Wenbo Ge[3], Josie E. Parker [6], Steven L. Kelly [7], P. David Rogers[3] & Jarrod R. Fortwendel [2,5] ✉

Triazoles, the most widely used class of antifungal drugs, inhibit the biosynthesis of ergosterol, a crucial component of the fungal plasma membrane. Inhibition of a separate ergosterol biosynthetic step, catalyzed by the sterol C-24 methyltransferase Erg6, reduces the virulence of pathogenic yeasts, but its effects on filamentous fungal pathogens like *Aspergillus fumigatus* remain unexplored. Here, we show that the lipid droplet-associated enzyme Erg6 is essential for the viability of *A. fumigatus* and other *Aspergillus* species, including *A. lentulus*, *A. terreus*, and A. nidulans. Downregulation of *erg6* causes loss of sterol-rich membrane domains required for apical extension of hyphae, as well as altered sterol profiles consistent with the Erg6 enzyme functioning upstream of the triazole drug target, Cyp51A/Cyp51B. Unexpectedly, *erg6*-repressed strains display wild-type susceptibility against the ergosterol-active triazole and polyene antifungals. Finally, we show that *erg6* repression results in significant reduction in mortality in a murine model of invasive aspergillosis. Taken together with recent studies, our work supports Erg6 as a potentially pan-fungal drug target.

*Aspergillus fumig*atus is the most prevalent *Aspergillus* species that causes invasive aspergillosis (IA), a life-threatening fungal infection with high mortality rates up to 40%–50%[1]. With the increasing numbers of patients having immune defects, *Aspergillus*-related infections have become an important public health concern[2]. Currently, there are only three available classes of antifungal compounds for the treatment of IA (i.e., triazoles, polyenes, and echinocandins), all targeting essential components of the fungal cell membrane or cell wall[3]. Unfortunately, the clinical efficacy of these antifungal classes is hampered by host toxicity, side effects, and poor bioavailability to some extent. Moreover, the global emergence of resistance, especially to the triazole class, makes their clinical application for long-term treatment more complicated[2,4].

Sterols are functional and constructional components residing in the plasma membrane and are responsible for the cell membrane permeability, fluidity, and stability[5]. Ergosterol (C28 sterol) is a specific sterol found in fungi and protozoa, whereas mammalian cells synthesize cholesterol (C27 sterol) as the major membrane sterol[6].

[1]Graduate Program in Pharmaceutical Sciences, College of Graduate Health Sciences, University of Tennessee Health Science Center, Memphis, TN, USA. [2]Department of Clinical Pharmacy and Translational Science, College of Pharmacy, University of Tennessee Health Science Center, Memphis, TN, USA. [3]Department of Pharmacy and Pharmaceutical Sciences, St. Jude Children's Research Hospital, Memphis, TN, USA. [4]Integrated Program in Biomedical Sciences, College of Graduate Health Sciences, University of Tennessee Health Science Center, Memphis, TN, USA. [5]Department of Microbiology, Immunology, and Biochemistry, College of Medicine, University of Tennessee Health Science Center, Memphis, TN, USA. [6]Molecular Biosciences Division, School of Biosciences, Cardiff University, Cardiff, Wales, UK. [7]Institute of Life Science, Swansea University Medical School, Swansea, Wales, UK. ✉e-mail: jfortwen@uthsc.edu

Because of the uniqueness and essentiality of ergosterol for fungal organisms, disturbing fungal ergosterol homeostasis is widely considered a promising strategy for novel antifungal development[7]. Besides triazoles and polyenes, statins and allylamines are two classes of inhibitors targeting ergosterol biosynthesis or ergosterol directly[8,9]. Ergosterol and cholesterol are both sterols with similar four-ring structure harboring a hydroxyl group at C-3 and an unsaturated bond at C-5,6. The distinguishing feature between ergosta-type and cholesta-type sterols is that ergosta-type sterols contain a methyl group at C-24 on the side chain. The addition of this methyl group is catalyzed by the sterol C-24 methyltransferase enzyme, encoded by the *erg6* gene in fungi[10]. Although many proteins involved in the fungal ergosterol biosynthesis pathway have orthologs in mammalian cholesterol biosynthetic processes, Erg6 is one of the three specific enzymes, also including Erg4 and Erg5, that are absent in humans[11].

Among the organisms studied to date, yeast and filamentous fungi appear to share conserved early and late enzymatic steps of the ergosterol biosynthesis pathway. However, after the formation of the first sterol-type intermediate, lanosterol, the pathway bifurcates into one of two paths. In budding yeast, like *Saccharomyces cerevisiae*, lanosterol is catalyzed to 4,4-dimethylcholesta-8,14,24-trienol by the triazole-target gene, Erg11[5]. As for *A. fumigatus*, eburicol is the preferred substrate of the Erg11 orthologs, Cyp51A and Cyp51B, and eburicol is generated from lanosterol by the activity of Erg6[12]. Therefore, although Erg6 represents one of the late enzymatic steps for ergosterol biosynthesis in *S. cerevisiae*, its substrate specificity for lanosterol makes it an early enzymatic step for organisms like *A. fumigatus*. Erg6 catalyzes a methyl addition to C-24 by the way of an S-adenosylmethionine (SAM)–dependent transmethylation and shifts a double bond to produce a C-24(28)-methylene structure with high substrate specificity[13]. Erg6 has been genetically characterized in multiple single-celled yeast, including *S. cerevisiae*, *Kluyveromyces lactis*, *Candida glabrata*, *Candida albicans*, *Cryptococcus neoformans*, and *Pneumocystis carinii*[14–18]. Deletion of *erg6* is not lethal for these fungi. However, *erg6* loss-of-function mutations in these organisms cause alteration of drug susceptibility and defective growth phenotypes related to membrane integrity and permeability. Recent exciting studies have shown that loss of *C. albicans* Erg6 activity, either through genetic downregulation of *ERG6* gene transcription or through pharmacologic inhibition of Erg6 protein activity with a novel small molecule, blocks the pathogenic yeast-to-hyphae transition and significantly reduces virulence[19,20]. Studies of Erg6 in filamentous fungi are relatively very limited. Recently, Erg6 has been characterized in *Mucor lusitanicus*. Unlike yeast genomes that encode only one copy of Erg6, *M. lusitanicus* possesses three copies, referred to Erg6A, Erg6B, and Erg6C[21]. Erg6B plays a critical role in ergosterol biosynthesis. Deletion of *erg6B* compromises ergosterol production, growth ability, antifungal resistance and virulence, and double deletion together with *erg6A* or *erg6C* is lethal for *M. lusitanicus*[21].

Although *A. fumigatus* encodes two putative sterol C-24 methyltransferases, designated as Erg6 and Smt1, in this study, we report that only loss of *erg6* generates significant phenotypes. Strikingly, we find that *erg6* is essential for *A. fumigatus* viability in vitro and for disease establishment in a murine model of IA. We also show that *erg6* orthologs are essential across multiple *Aspergillus* species. Repression of *A. fumigatus erg6* expression in a conditional mutant blocks ergosterol biosynthesis resulting in abundant accumulation of lanosterol, the proposed substrate of Erg6. Surprisingly, the downregulation of *erg6* does not drive significant changes in triazole or polyene susceptibility profiles. This result is contrary to *erg6* mutants in other fungal species. Taken together, our data support inactivation of Erg6 as a possible therapeutic approach for fungal infection.

## Results

### Erg6 is indispensable for *A. fumigatus* viability

To identify putative *A. fumigatus* orthologs of *S. cerevisiae ERG6*, we performed a BLASTP analysis using the amino acid sequence of *S. cerevisiae* Erg6p (SGD: S000004467) against the *A. fumigatus* genome database[22]. Two putative protein-encoding loci, AFUB_099400 (EDP47339, 54.18% identity, 95% coverage) and AFUB_066290 (EDP50296, 29.49% identity, 55% coverage), which are designated as *erg6* and *smt1*, respectively, were identified. Alignment analysis showed that the putative *A. fumigatus* Erg6 and Smt1 proteins share 26.71% amino acid identity with each other. To explore the phylogenetic relationship of sterol C-24 methyltransferase in different fungi, the same analysis was performed in *A. lentulus*, *A. terreus*, *A. nidulans*, *S. cerevisiae*, *C. albicans*, *C. neoformans*, and *Neurospora crassa*. A phylogenetic tree was constructed based on full-length amino acid sequences using the maximum likelihood method (Fig. S1A). Remarkably, only a single sterol C-24 methyltransferase encoding gene was found in the yeast organisms analyzed, whereas the filamentous fungi analyzed each harbored at least two putative paralogs (Fig. S1A). The S-adenosylmethionine (SAM)-dependent methyltransferase enzymes display pronounced variability in sequence yet share a highly conserved structural fold[23]. *A. fumigatus* Erg6 and Smt1 proteins are predicted to harbor a conserved methyltransferase domain orchestrating SAM binding, with an identity of 71% and 37%, respectively, when aligned with methyltransferase domain from *S. cerevisiae* (Fig. S1B). Additionally, Erg6 of *A. fumigatus* and *S. cerevisiae* are both predicted to encode a conserved sterol methyltransferase C-terminal domain, which is responsible for selective substrate binding[23,24]. This conserved substrate-binding region was not found in *A. fumigatus* Smt1 (Fig. S1B).

To investigate the importance of sterol C-24 methyltransferase activity in *A. fumigatus*, we first attempted to generate null mutants of both *erg6* and *smt1*, completely replacing the open reading frames (ORFs) with a hygromycin selection cassette, using a highly efficient CRISPR/Cas-9 gene editing technique (Fig. S2A)[25]. No Δ*erg6* transformant was obtained after several transformation attempts, whereas Δ*smt1* mutants were successfully generated. These results implied a differential requirement for the two putative sterol C-24 methyltransferases in *A. fumigatus*, with the *erg6* homolog potentially being essential. Phenotypic analysis of the Δ*smt1* mutant revealed no differences in colony growth or morphology when compared to the control strain, suggesting that either Smt1 plays a minimal role in ergosterol biosynthesis or that Erg6 activity is able to compensate for the loss of *smt1* (Fig. S4A and S5A). To generate a hypomorphic allele of *erg6* for further study, we next constructed a pTetOff-*erg6* mutant in which the endogenous *erg6* promoter was replaced by a tetracycline-repressible promoter (Fig. S2B)[26,27]. Although this genetic manipulation resulted in a 4-fold ($\log_2$) increase in *erg6* expression in the absence of doxycycline, the presence of only 0.5 μg/ml doxycycline in the culture media generated a 4-fold ($\log_2$) reduction in gene expression (Fig. 1A). Importantly, the pTetOff-*erg6* mutant displayed growth and colony morphology identical to the parental strain when cultured in the absence of doxycycline, suggesting that the basal upregulation of *erg6* expression in the absence of doxycycline does not alter basic growth of *A. fumigatus* (Fig. 1B, left panel). However, growth and germination were significantly inhibited in the pTetOff-*erg6* strain in the presence of increasing doxycycline concentrations (Fig. 1B, C). As low as 0.5 μg / ml doxycycline almost completely prevented colony development on solid agar (Fig. 1B). Analysis of submerged culture demonstrated that the pTetOff-*erg6* strain exhibited a dose-dependent decrease in mycelial development in response to increasing doxycycline concentrations (Fig. 1C).

Together, these findings suggested that *erg6* is likely essential for *A. fumigatus* viability. However, the pTetOff-*erg6* conidia were able to germinate and establish initial polarity in the highest concentrations of doxycycline tested (40 μg / ml) (Fig. 1C). Although this continued ability to germinate is likely due to "leakiness" of the doxycycline

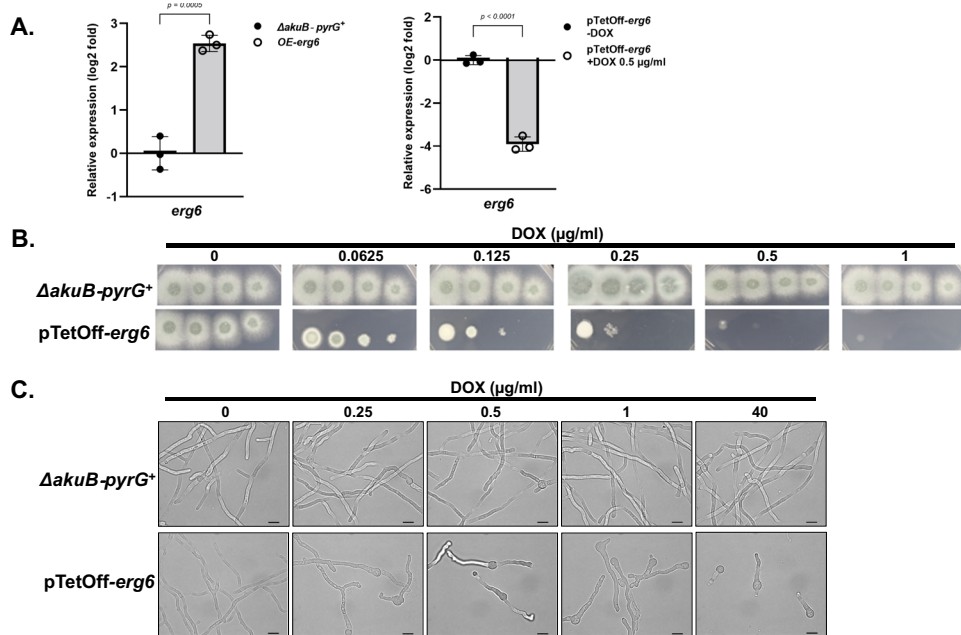

**Fig. 1 | Repression of *erg6* expression inhibits *A. fumigatus* growth in vitro.**
**A** The expression level of *erg6* in indicated conditions as analyzed by RT-qPCR. Mycelia were harvested after 16 h in liquid GMM at 37 °C, 250 rpm. Gene expression was normalized to the reference gene, *tubA*, and data presented relative to control group as mean ± SD of log₂ fold change. *n* = 3 independent experiments. Two-tailed Student t-test was used for statistical analysis. **B** Spot-dilution assays were performed on GMM agar plates with the parental and pTetOff-*erg6* strains in the indicated doxycycline (DOX) levels. For all assays, suspension aliquots of 5 μl containing 50,000, 5,000, 500, and 50 total conidia were inoculated and plates were incubated at 37 °C for 48 h. **C** Microscopic images of the parental and pTetOff-*erg6* strains after 16 h of exposure to the indicated doxycycline levels in static GMM culture at 37 °C. Microscopy was performed on Nikon NiU with bright field settings. Scale bar = 10 μm.

repressible promoter system, these findings could also imply that Erg6 activity is only essential for growth and viability post-polarity establishment. To further test if *erg6* is differentially essential for pre- or post-germination viability, we next constructed a pTetOn-*erg6* mutant (Fig. S2B) that should require the presence of doxycycline for *erg6* expression[28]. Importantly, the pTetOn-*erg6* mutant behaved as expected in culture, with colony development only occurring upon the addition of doxycycline to the media (Fig. S6). Both pTetOn-*erg6* and pTetOff-*erg6* strains were employed in live-cell staining assays using the fluorescent marker 5-carboxyfluorescein diacetate (CFDA)[29]. Parental and pTetOn-*erg6* conidia were cultured in GMM broth with or without 100 μg / ml doxycycline for 16 hours, followed by CFDA staining to quantitatively measure the viability of germlings. Germlings that were either fully or only partly CFDA-labeled were counted as viable cells. The cultures were limited to 16 hours of incubation to allow unambiguous detection of live *vs.* dead (i.e., stained *vs.* unstained) fungal elements. As shown in Fig. 2B, upper panel, without doxycycline as an inducer, no polarity was observed in pTetOn-*erg6* with rare viable conidia staining positive with CFDA. In the presence of 100 μg / ml doxycycline, the pTetOn-*erg6* mutants displayed germination and viability rates comparable to the parental background strains (Fig. 2A, left panel, and 2B, upper panel). Thus, our data confirm that *erg6* is required for *A. fumigatus* viability, beginning with the earliest stages of growth.

Similar results were achieved when using the pTetOff-*erg6* mutant. Parental and pTetOff-*erg6* conidia were cultured in GMM broth with or without 4 μg/ml doxycycline for 16 hours. Nearly all germlings for the parental and pTetOff-*erg6* strains cultured in doxycycline-free conditions were positively fluorescently labeled with CFDA, indicating ~100% viability in nonrepressive conditions (Fig. 2A, right panel, and 2B, lower panel). In the presence of 4 μg/ml doxycycline, whereas the parental strain remained unaffected, the pTetOff-*erg6* displayed a sharp decrease in the CFDA-labeled population of germlings with only 20% positivity (Fig. 2A, right panel). To further

confirm *erg6* deficiency is lethal for *A. fumigatus*, conidia of parental and pTetOff-*erg6* strains were cultured to the germling stage in GMM broth without doxycycline and subsequently inoculated onto GMM agar containing increasing concentrations of doxycycline. As shown in Fig. 2C, similar to the hyphal growth inhibition exhibited by culturing conidia on doxycycline-impregnated agar plates, pre-formed germlings of the pTetOff-*erg6* mutant were entirely growth inhibited with as little as 0.5 μg/ml doxycycline. Taken together, these findings demonstrate that *erg6* deficiency is lethal for *A. fumigatus*.

To determine if the essentiality of *erg6* was not only specific to *A. fumigatus* but may instead be generalizable across *Aspergillus* species, we next generated tetracycline-repressible promoter replacement mutants targeting the *erg6* orthologs of *A. lentulus*, *A. terreus*, and *A. nidulans* through the CRISPR/Cas-9 technique. Erg6 orthologs were retrieved from the most similar alignments in BLASTP analysis against the genome databases of the respective *Aspergillus* species using the amino acid sequence of *S. cerevisiae* *erg6* (SGD: S000004467) as a query sequence. Replacement of the endogenous *erg6* promoter with the pTetOff system resulted in no significant changes in growth versus parental controls for both *A. nidulans* and *A. lentulus* under doxycycline-free conditions (Fig. S4B and C). For *A. terreus*, pTetOff promoter replacement resulted in a significant decrease in colony diameter under doxycycline-free conditions when compared to the parental strain after 48 and 72 hrs of culture (Fig. S4D). Importantly, as we noted for *A. fumigatus*, there was a clear negative correlation between increasing doxycycline concentrations and colony establishment on agar plates among all three additional filamentous *Aspergillus* species (Fig. 3). Thus, *erg6* is essential in multiple *Aspergillus* species.

## *smt1* is not a functionally redundant paralog of *erg6*

We next sought to examine the functional relationship between *erg6* and the predicted paralog, *smt1*. To address whether overexpression of *smt1* could rescue *erg6* repression, we constructed a pTetOff-*erg6*

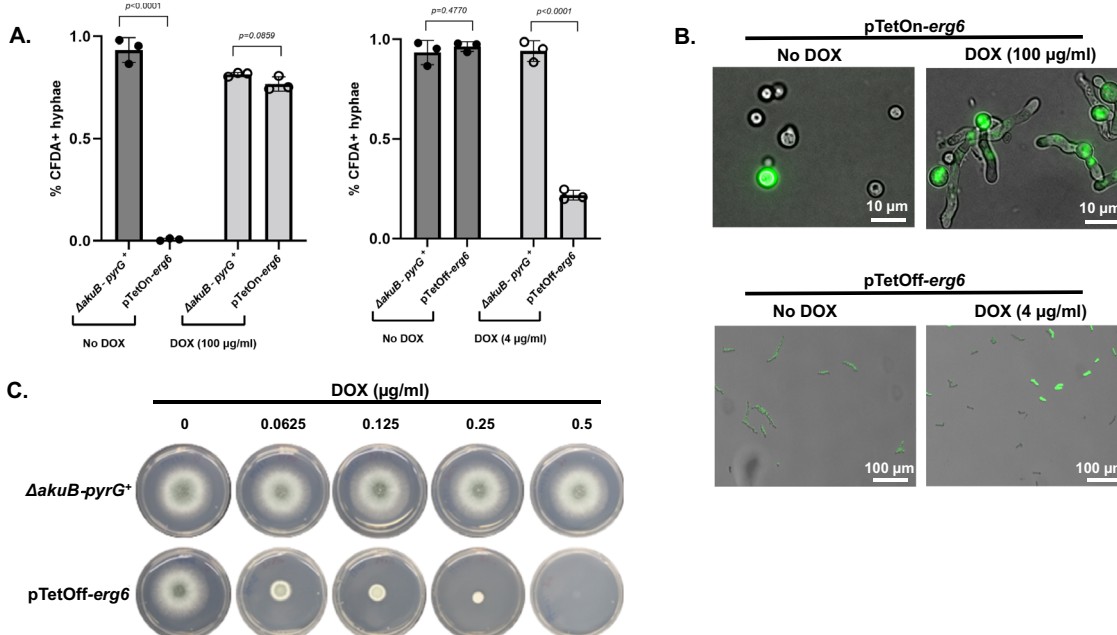

**Fig. 2 | Erg6 is essential for *A. fumigatus* viability. A** Viability rates of pTetOn-*erg6* and pTetOff-*erg6* in the indicated doxycycline (DOX) levels using a CFDA staining assay. Hyphae were harvested after GMM culture for 16 h at 30 °C and subsequently stained with 50 µg/ml CFDA for 1 h. Microcolonies that showed bright a green signal were manually enumerated as viable. *n* = 3 independent experiments. More than 100 microcolonies were measured in each assay. Data is depicted as the mean ± SD. Two-tailed Student *t*-tests were used for statistical analysis. ns = not significant. **B** Representative images of pTetOn-*erg6* and pTetOff-*erg6* microcolonies stained with the live-cell dyes CFDA treated with indicated doxycycline levels. **C** Conidia of the parental and pTetOff-*erg6* strains were initially grown in GMM broth without doxycycline treatment for 8 h to allow the formation of germlings. Subsequently, germling aliquots of 10 µl were transferred to fresh GMM agar plates supplemented with the indicated concentration of doxycycline, and plates were incubated for an additional 48 h.

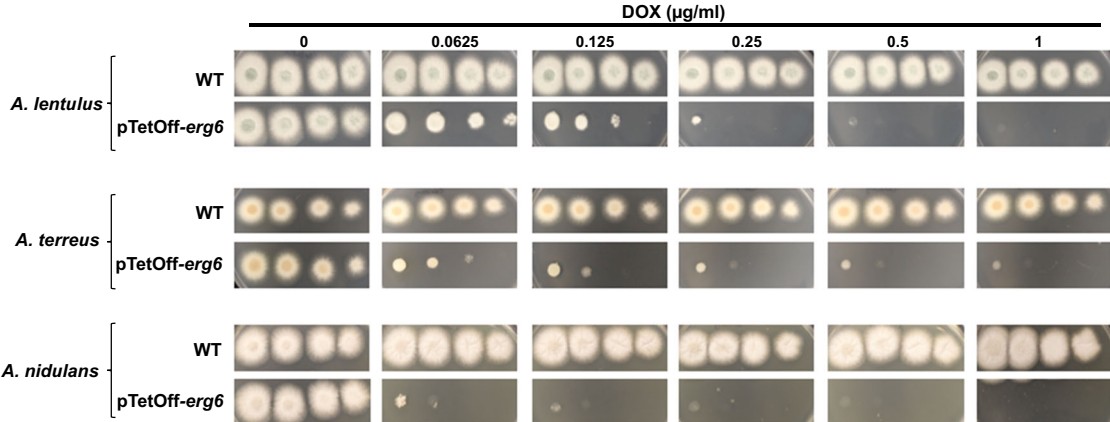

**Fig. 3 | Erg6 is essential across *Aspergillus* species.** Colony morphology of parental and pTetOff-*erg6* strains of *A. lentulus*, *A. terreus*, and *A. nidulans* in the presence of the indicated concentrations of doxycycline (DOX). Spot-dilution culture was performed as described in Fig. 1B. For *A. lentulus* and *A. terreus*, conidia were inoculated onto GMM plates, whereas *A. nidulans* conidia were cultured on GMM supplemented with 5% yeast extract.

mutation in a *smt1* overexpression background. The *smt1* endogenous promoter was first replaced with the strong p*HspA* promoter[30] through CRIPSR/Cas9-mediated gene targeting to generate strain OE-*smt1* (Fig. S2C). Although this promoter replacement generated a ~6-fold (log$_2$) upregulation of *smt1* expression, growth and colony development were unaffected (Fig. S4A, S5A, and S5B). The pTetOff-*erg6* promoter construct was then integrated in the OE-*smt1* genetic background. When these mutants were employed in spot-dilution assays, constitutive *smt1* overexpression driven by the p*HspA* promoter was not able to promote colony development when *erg6* was downregulated by doxycycline addition (Fig. 4A). Further, using RT-qPCR to measure *erg6* and *smt1* expression levels, we found that the expression of neither *erg6* nor *smt1* was responsive to loss of the other (Fig. S5C). To

further rule out the possibility that *smt1* compensates for *erg6* deficiency, a pTetOff-*erg6* mutation was constructed in the Δ*smt1* genetic background. As shown in Fig. 4B, deletion of *smt1* was not found to exacerbate the loss of viability when *erg6* expression was repressed by the addition of exogenous doxycycline. An almost complete lack of colony development was evident at 0.5 µg/ml doxycycline as was seen in pTetOff-*erg6* strains expressing *smt1* (Fig. 1C). Therefore, *smt1* does not appear to be a functional paralog of *erg6*.

**Repression of *erg6* results in loss of membrane ergosterol and altered sterol profiles**
Although *smt1* appeared to play little-to-no role in ergosterol biosynthesis to support growth, we next sought to ensure a conserved

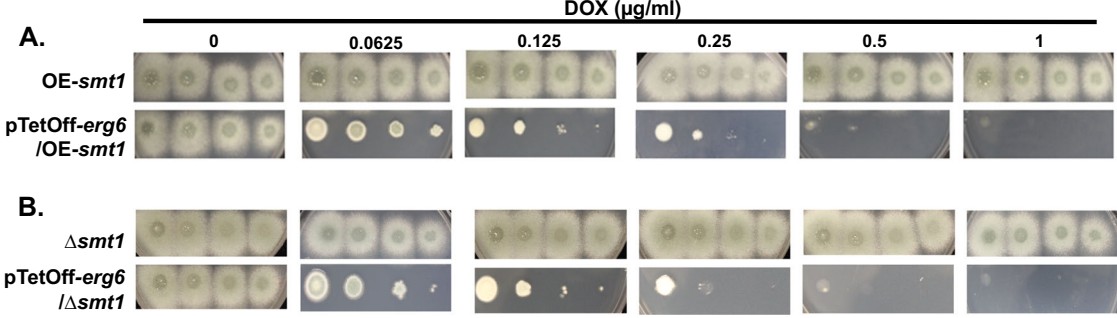

**Fig. 4 | The putative paralog, *smt1*, shows no functional redundancy with *erg6*.** Spot-dilution assays of pTetOff-*erg6* mutants constructed in the Overexpression (OE)-*smt1* (**A**) or Δ*smt1* (**B**) genetic backgrounds. Culture conditions were as described in Fig. 1B. DOX = doxycycline.

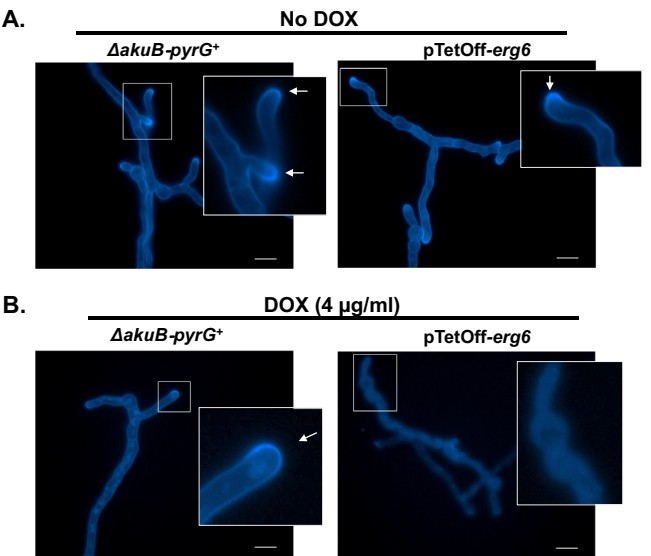

**Fig. 5 | Repression of Erg6 expression alters ergosterol distribution in *A. fumigatus* hyphae.** Mycelia of the parental and pTetOff-*erg6* strains were grown in GMM broth without doxycycline (DOX) (**A**) and with 4 µg/ml doxycycline (**B**) for 16 h at 30 °C. Hyphae were subsequently stained with 25 µg/ml filipin for 5 min. Fluorescent images were captured using DAPI filter settings. White arrows indicate sterol-rich plasma membrane domains (SRDs). Scale bar = 10 µm. Microscopy experiments were completed three times independently with similar results.

role for *erg6* in this important pathway for *A. fumigatus*. In filamentous fungi, ergosterol is known to accumulate in hyphal tips in structures called sterol-rich plasma membrane domains (SRDs), which have been validated to be essential cellular machinery involved in the maintenance of growth polarity[31]. Because *erg6* is a putative ergosterol biosynthesis pathway component and loss of *erg6* expression causes severe hyphal growth impairment, we hypothesized that *erg6* downregulation would result in the loss of ergosterol accumulation at hyphal tips, loss of total cellular ergosterol, and accumulation of the putative Erg6 substrate, lanosterol[12]. To test this, we first stained hyphae of the parental and pTetOff-*erg6* strains with the sterol dye, filipin, which is widely used in filamentous fungi to visualize SRDs[31]. As shown in Fig. 5A, B, filipin staining of the parental strain revealed concentrated fluorescence at the hyphal tips with and without doxycycline treatment, forming a cap-like pattern structure as indicated by the white arrows. In the absence of doxycycline, the pTetOff-*erg6* mutant behaved similarly (Fig. 5A, right panel). However, in the pTetOff-*erg6* doxycycline-treated cultures, the filipin staining pattern was completely disrupted with diminished hyphal staining and a loss of specific hyphal tip accumulation (Fig. 5B, right panel). To measure

sterol profiles quantitatively, total sterols were derivatized to trimethylsilyl ethers and analyzed using gas chromatography (GC)-mass spectrometry (MS) in both strains under increasing doxycycline concentrations. Sterol profiles of the parental strain were comparable in the presence or absence of doxycycline treatment, with ergosterol accounting for nearly 90% of total sterol and the Erg6 substrate, lanosterol, only accounting for ~0.6% (Table 1). As expected, the pTetOff-*erg6* mutant displayed sterol profiles similar to the parent strain when no doxycycline was added to the culture medium (Table 1). In contrast, among the pTetOff-*erg6* doxycycline treatment groups, the total ergosterol content decreased by almost 50% and lanosterol accumulated significantly to the second-most abundant measured sterol to nearly 40% of the total sterols present. Although undetectable in the parent strain, several cholesta-type intermediates, including cholesta-5,7,22,24-tetraenol, cholesta-5,7,24-trienol, 4,4-dimethyl cholesta-dienol and cholesta-dienol, each accounted for less than 4% of total sterols in the doxycycline-treated pTetOff-*erg6* mutant. Taken together, these findings further confirm a conserved role for *A. fumigatus erg6* in ergosterol biosynthesis, specifically at the lanosterol-to-eburicol conversion step.

### *A. fumigatus* Erg6 localizes to lipid bodies
The Erg6 sterol C-24 methyltransferase homolog has been reported to localize to lipid droplets and to the endoplasmic reticulum of the model yeast, *S. cerevisiae*[32,33]. To examine if the localization of Erg6 in a filamentous pathogenic mold, like *A. fumigatus*, is conserved, we performed localization studies using strains expressing an Erg6-enchanced Green Fluorescent Protein (eGFP) chimera. Employing both the parental and pTetOff-*erg6* backgrounds, a construct was designed to fuse *egfp* to the 3' end of *erg6*, such that *erg6-gfp* expression would be controlled by the native *erg6* promoter in the parental background and by the TetOff promoter in the pTetOff-*erg6* background (Fig. S2D). Phenotypic assays indicated that the Erg6-GFP mutants were functionally normal and that the pTetOff-*erg6-gfp* strain was as equally responsive as the nonchimeric mutant to doxycycline-mediated *erg6* repression (Fig. 6A). These results indicated that the GFP-fusion had no detrimental effects on Erg6. Fluorescent microscopic observation revealed that, regardless of doxycycline presence, Erg6-GFP displayed a punctate localization pattern distributed evenly throughout the mycelia of the *erg6-gfp* strain (Fig. 6B, upper panels). Notably, we observed that the Erg6-GFP signal in pTetOff-*erg6-gfp* strain cultured without doxycycline was much stronger than that of the *erg6-gfp* strain (Fig. 6B, lower left panel). These protein-level findings are consistent with our previous data showing basal overexpression of *erg6* when under the control of the TetOff promoter and cultured in the absence of doxycycline (Fig. 1A, left panel). Regardless of protein abundance, Erg6-GFP localization remained confined to punctate structures dispersed throughout hyphae of the pTetOff-*erg6-gfp* strain in the absence of doxycycline. In contrast, the Erg6-GFP signal of the

**Table 1 | Alteration of total sterol composition in response to *erg6* repression**

| Type of sterol | ΔakuB-pyrG+ | | | pTetOff-erg6 | | |
|---|---|---|---|---|---|---|
| | NT | 0.5 µg/mL Dox | 2 µg/mL Dox | NT | 0.5 µg/mL Dox | 2 µg/mL Dox |
| Ergosta-5,8,22,24(28)-tetraenol | 0.4 ± 0.3 | 0.4 ± 0.1 | 0.6 ± 0.2 | 1.4 ± 0.4 | 0.3 ± 0.3 | 0.6 ± 0.5 |
| Cholesta-5,7,24-trienol | | | | | 1.4 ± 0.7 | 0.7 ± 0.7 |
| Ergosta-5,8,22-trienol | 0.9 ± 0.2 | 1.2 ± 0.6 | 0.8 ± 0.2 | 1.4 ± 0.4 | | |
| Cholesta-dienol | | | | | 1.1 ± 0.4 | 0.7 ± 0.7 |
| Ergosterol | **90.4 ± 1.3** | **90.2 ± 2.2** | **90.5 ± 1.6** | **88.6 ± 2.8** | **47.4 ± 5.5** | **43.4 ± 5.0** |
| Ergosta-5,7,22,24(28)-tetraenol | | | | | 0.3 ± 0.4 | |
| Cholesta-5,7,22,24-tetraenol | | | | | **2.5 ± 0.4** | **3.6 ± 0.8** |
| 4,4-Dimethyl cholesta dienol | | | | | **2.6 ± 0.1** | 1.9 ± 1.7 |
| Ergosta-5,7,24(28)-trienol | 1.1 ± 0.4 | 0.7 ± 0.5 | 1.1 ± 0.7 | 0.4 ± 0.7 | | |
| Ergosta-5,7-dienol | 1.3 ± 0.2 | 1.3 ± 0.2 | 1.0 ± 0.2 | 0.9 ± 0.9 | | |
| Episterol [Ergosta-7,24(28)-dienol] | 0.8 ± 0.2 | 0.7 ± 0.2 | 0.7 ± 0.1 | 0.8 ± 0.1 | | |
| Lanosterol | 0.6 ± 0.1 | 0.5 ± 0.2 | 0.6 ± 0.3 | 0.5 ± 0.5 | **39.8 ± 6.0** | **42.7 ± 4.4** |
| 4-Methyl ergosta-8,24(28)-dienol | 0.8 ± 0.1 | 0.8 ± 0.1 | 0.7 ± 0.1 | 1.0 ± 0.2 | | |
| Eburicol | 1.0 ± 0.2 | 1.0 ± 0.1 | 1.0 ± 0.1 | 1.5 ± 0.5 | 1.3 ± 1.2 | **2.4 ± 2.2** |
| 4,4-Dimethyl ergosta-8,24(28)-dienol | 1.4 ± 0.1 | 1.4 ± 0.2 | 1.3 ± 0.5 | 1.9 ± 0.9 | | |

Data presented as means for 3 replicates with standard deviation are percentage of the total sterol composition. Values of >2% of the total sterol composition are shown in bold. *NT* No Treatment.

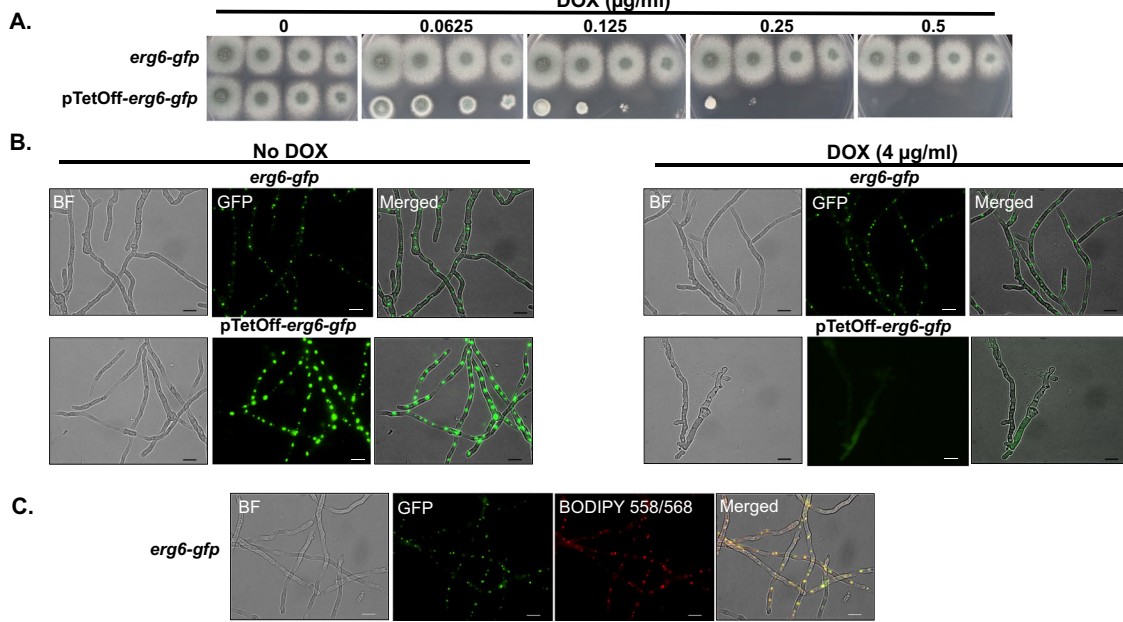

**Fig. 6 | Erg6 localizes to lipid droplets in *A. fumigatus* hyphae. A** Spot-dilution cultures, performed as described in Fig. 1B, indicate that fusion of *egfp* to the 3' end of *erg6* in either the parent or pTetOff-*erg6* background does not negatively affect *erg6* function. Note similarities to growth in untagged strains (Fig. 1B). **B** Mature mycelia were developed in GMM broth using the indicated doxycycline (DOX) concentrations for 16 h at 37 °C. Fluorescent images were captured using GFP filter settings. **C** Co-localization of Erg6-GFP to *A. fumigatus* lipid droplets using the droplet marker, BODIPY 558/568. Conidia of the erg6-gfp strain were cultured to mature hyphal development and subsequently stained with 1 µg/ml BODIPY 558/568 C12 for 20 min at room temperature. Images were captured using GFP and TRITC filter settings, respectively. Scale bar = 10 µm. Microscopy experiments were completed three times independently with similar results.

pTetOff-*erg6-gfp* mutant was significantly reduced in the presence of doxycycline, confirming the loss of Erg6 at the protein level when *erg6* gene expression was repressed (Fig. 6B, lower right panel). To demonstrate that the punctate localization of Erg6 overlapped with lipid droplets directly, we next stained for lipid droplets in the *erg6-gfp* strain using the lipophilic fluorescent dye BODIPY 558/568 C$_{12}$, a specific tracer of lipid trafficking[34,35]. As detected by fluorescent microscopy, the GFP-labeled puncta overlapped with the red BODIPY staining completely (Fig. 6C). This finding indicated that Erg6-GFP co-

localized with lipid droplets in actively growing hyphae. Therefore, our data demonstrate that, similar to *S. cerevisiae*, Erg6 localizes to *A. fumigatus* lipid droplets.

### Repression of *erg6* does not alter susceptibility to ergosterol-targeted antifungals

Multiple classes of currently available antifungal drugs target ergosterol or the ergosterol biosynthesis pathway to destabilize cell membrane integrity and function[36]. As *erg6* is required for the biosynthesis

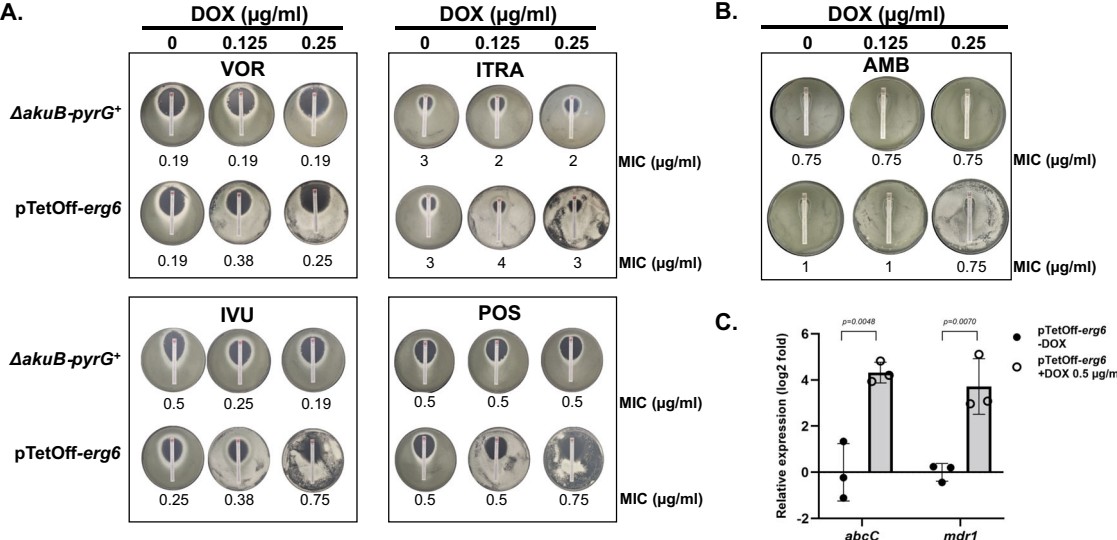

**Fig. 7 | Downregulation of *erg6* does not alter antifungal susceptibility in *A. fumigatus*.** Strip-diffusion MIC assays of the parental and pTetOff-*erg6* strains were carried out under the indicated doxycycline (DOX) concentrations. Conidia (2 × 10⁶) suspended in 0.5 ml sterile water were spread evenly over GMM plates and allowed to dry. Commercial test strips embedded with voriconazole, itraconazole, isavuconazole, posaconazole (**A**) or amphotericin B (**B**) were applied and plates were incubated for 48 h at 37 °C. The resulting MIC values are indicated at the bottom of each plate image. AMB, amphotericin B, VOR, voriconazole, ITRA, itraconazole, IVU, isavuconazole, POS, posaconazole. (**C**). The expression levels of *abcC* and *mdr1* under the indicated conditions were analyzed by RT-qPCR. Gene expression was normalized to the reference gene, *tubA*, and data is presented relative to control group as mean ± SD of log₂ fold change. *n* = 3 independent experiments. Two-tailed Student t-test was used for data analysis.

of ergosterol in yeast species, *erg6* gene mutation has been described as resulting in *Candida* and *Cryptococcus* yeast cells with decreased ergosterol content and increased resistance to the ergosterol-binding antifungal drug, amphotericin B[17,37]. In contrast, the *C. neoformans erg6* null mutant has been shown to be hypersusceptible to the triazole antifungals, a class of lanosterol-14-α-demethylase inhibitors[17]. To first explore if triazole stress impacted *erg6*, we assayed the expression of *erg6* at the transcriptional and protein level in response to voriconazole. Culture of *A. fumigatus* in the presence of sub-MIC voriconazole (0.125 μg/ml) generated a ~3-fold (log2) increase in *erg6* gene expression (Fig. S7A). This increased gene expression translated to the protein level, as the culture of the *erg6-gfp* strain under the same conditions resulted in upregulation of the lipid droplet-associated GFP signal (Fig. S7B). These results suggested that *erg6* expression is triazole-stress responsive. To explore whether *erg6* repression alters antifungal susceptibility in *A. fumigatus*, we carried out MICs assays in the parental strain and pTetOff-*erg6* mutant by strip-diffusion assays. So that sufficient mycelia were obtained for the pTetOff-*erg6* mutant under repressive conditions to accurately monitor the zone-of-inhibition, we utilized the sub-lethal concentrations of 0.125 and 0.25 μg/ml doxycycline embedded GMM agar plates in combination with voriconazole, isavuconazole, itraconazole, posaconazole, and amphotericin B strips. Unexpectedly, after 48 hours of culture, no significant difference in MIC (2-fold or more change) was noted under any condition (Fig. 7A, B). The results of these strip-diffusion tests were consistent with broth micro-dilution antifungal susceptibility testing (Fig. S8A–E). To verify whether the susceptibility profiles under *erg6*-repressed conditions might be affected by other factors, we also measured the expression of two efflux pump genes, *abcC* and *mdr1*, associated with resistance to triazoles[38]. Whereas doxycycline treatment had no influence on the expression of either efflux pump in the parental strain, surprisingly, RT-qPCR analysis revealed that *abcC* and *mdr1* were overexpressed 3- to 5-fold (log2) under *erg6* repression conditions in the pTetOff-*erg6* mutant, compared to the no-doxycycline control (Fig. 7C). Therefore, it is possible that increased efflux pump activity under *erg6* repression might counterbalance the accumulation of antifungals in fungal cells, especially for the triazoles for which efflux is a characterized resistance mechanism.

## Expression of *erg6* is essential for development of invasive aspergillosis

Our in vitro results demonstrate that *erg6* is required for viability of *A. fumigatus*. To determine the effect of *erg6* repression in vivo, we next compared survival rates following infection with the parental strain and pTetOff-*erg6* mutant with or without doxycycline administration in a chemotherapeutically immune-suppressed mouse model of invasive aspergillosis. As a further measure of the requirement for *erg6* expression for in vivo viability of *A. fumigatus*, we also utilized the pTetOn-*erg6* mutant in the same disease model without doxycycline administration. The TetOff system employed here has been validated to regulate the expression of target genes in a murine-invasive pulmonary aspergillosis model[39,40]. Previous studies reported that the efficacy of the TetOff system is affected by the starting time of doxycycline administration in vivo, and administration beginning 1 day prior to infection yields effective target gene regulation[39]. However, it is well documented that intense doxycycline regimens are toxic to mice, resulting in weight loss and lethargy symptoms similar to infection[40]. Therefore, in this study, we applied doxycycline twice per day (100 mg/kg, gavage) beginning 3 days prior to infection, switching to once per day at day 4 post-infection and continuing to the end of the experiment. Mice (*n* = 10 per group) were chemotherapeutically immune suppressed with cyclophosphamide and triamcinolone acetonide as described in Materials and Methods and intranasally inoculated with the indicated strains on day 0. Mice infected with the parental strain (Δ*akuB-pyrG*⁺) experienced 100% mortality by day 6 post-inoculation and doxycycline administration did not affect this outcome (Fig. 8A). In contrast, whereas mice infected with the pTetOff-*erg6* mutant displayed similar mortality to the parental strain in the absence of doxycycline, pTetOff-*erg6* infected mice receiving doxycycline experienced only 30% mortality (Fig. 8A). Strikingly, mice infected with the pTetOn-*erg6* mutant in the absence of doxycycline administration displayed 100% survival through the end of the experiment (Fig. 8A). To confirm that the survival results

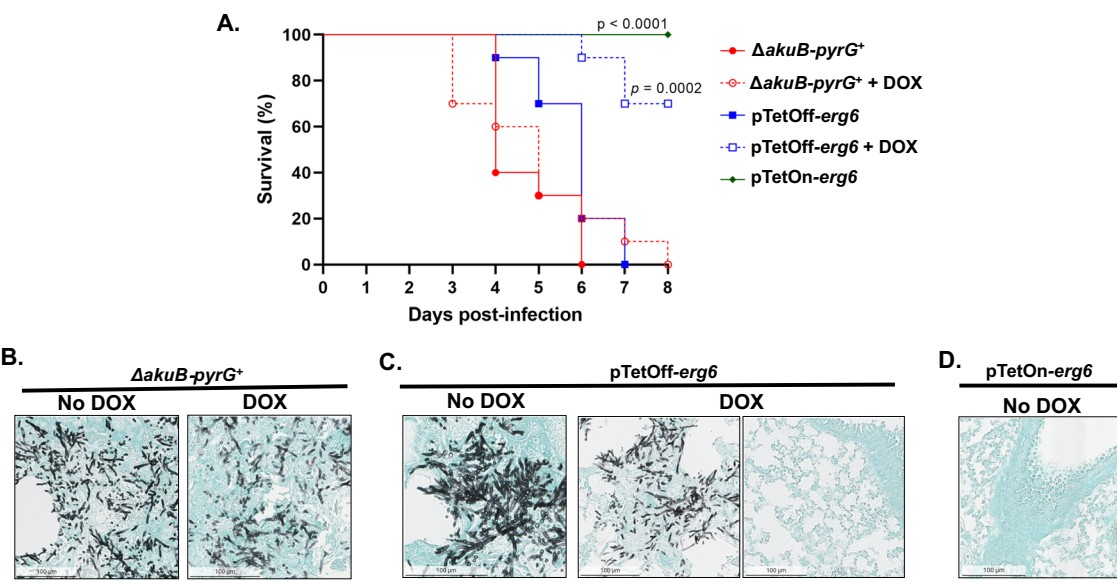

**Fig. 8 | Repression of *erg6* reduces virulence in a murine model of invasive aspergillosis. A** Survival analysis of mice infected with the indicated strains with or without doxycycline (DOX). Mice (*n* = 10/group) were immune suppressed chemotherapeutically using cyclophosphamide and triamcinolone acetonide as described in Material and Methods and inoculated with $1 \times 10^6$ conidia. For the doxycycline treatment arms, doxycycline (100 mg/kg) was supplied by gavage twice a day from Day −3 to Day 3 then once a day from Day 4 to the end of study. Data were analyzed by Mantel-Cox log-rank test comparing doxycycline-treated arms to their no doxycycline control. The pTetOn-*erg6* group was compared to the Δ*akuB-pyrG*⁺ group. **B**−**D** Representative images of Grocott's Methenamine Silver-stained lung tissue from mice of **A**. Lungs (*n* = 3 / group) were harvested 3 days post-infection. Hyphae are stained black by GMS.

were reflective of fungal growth in vivo, mice (*n* = 3/group) were immune suppressed and infected in the same manner and lungs were removed at 3 days post-infection for histological examination. Lungs from all mice infected with the parental control strain displayed extensive fungal growth regardless of doxycycline administration (Fig. 8B). Similarly, in the absence of doxycycline administration, all lungs from mice infected with the pTetOff-*erg6* mutant displayed fulminant fungal growth (Fig. 8C). In contrast, lung tissue sections from one of three mice infected with the pTetOff-*erg6* mutant displayed no visible fungal growth with the other two remaining positive when doxycycline was administered to repress *erg6* expression (Fig. 8C). This result is supportive of the survival study, where this experimental arm displayed significantly reduced mortality but did not reach 100% survival. To ensure that *erg6* expression is absolutely required for the viability of *A. fumigatus* in vivo, we also examined lung tissue sections of mice infected with conidia from the pTetOn-*erg6* mutant in the absence of doxycycline administration. These lung sections revealed no visible fungal growth by GMS staining (Fig. 8D). Therefore, e*rg6* is essential for in vivo viability of *Aspergillus* during the initiation of invasive aspergillosis.

## Discussion

Ergosterol is an essential sterol component of the fungal plasma membrane and is involved in numerous architectural and biological functions, such as membrane integrity, fluidity, and permeability (reviewed in[5,41]). A robust body of literature has demonstrated that disturbed ergosterol homeostasis results in membrane dysfunction and even cell death[42]. While ergosterol biosynthesis has been well explored in yeast, studies in filamentous fungi have been relatively limited. In this study, we characterized the sterol C-24 methyltransferase encoding gene, *erg6*, as an essential gene in *A. fumigatus* and analyze roles for this gene in growth, ergosterol biosynthesis, drug resistance and establishment of infection.

Some studies have defined *A. fumigatus* having only one copy of sterol C-24 methyltransferase, namely *erg6*[43,44], whereas others have described *smt1* orthologs as putative *erg6* paralogs[45]. Based on our analysis revealing *A. fumigatus* Smt1 to have 29.49% identity to *S. cerevisiae* Erg6 in amino acid sequence, it is possible that *smt1* is a paralog of *erg6* encoding sterol C-24 methyltransferase in *A. fumigatus*. Unlike yeast, it is common for the genomes of filamentous fungi to encode multiple paralogous genes in the ergosterol synthesis pathway. Gene duplications tend to generate functional redundancies and protections against negative effects of genetic mutations[46,47]. In our study, while both Erg6 and Smt1 are characterized as putative sterol C-24 methyltransferases, neither responded to the absence of the other in transcriptional level (Fig. S5C). Our findings mostly support the conclusion that either *smt1* is not a functional paralog of *erg6*, or *smt1* contributes little to C-24 methyltransferase activity in support of ergosterol biosynthesis in *Aspergillus*. For example, the Δ*smt1* mutant demonstrates a wild-type growth phenotype (Fig. S5A), *smt1* overexpression is unable to restore the phenotypic defects resulting from *erg6* deficiency (Fig. 4A), and loss of *smt1* does not cause exaggerated phenotypes in an *erg6* repressed mutant (Fig. 4B). Therefore, if Smt1 functions in ergosterol biosynthesis, we postulate that *erg6* acts as the predominant sterol C-24 methyltransferase and is able to compensate the loss of *smt1*. Our results clearly indicate that Smt1 does not contribute to cellular C24-methyltransferae activity to the extent that it can compensate for the loss of Erg6. Further functional experiments and sterol analyses still need to be performed to confirm the role of Smt1.

In this study, we validated that lipid droplet-localized Erg6 is essential for *A. fumigatus* survival in vitro and is required for fungal infection in vivo. As Δ*erg6* was unviable, we constructed tetracycline-regulatable *erg6* mutants as an alternative to explore the essentiality of Erg6. Without *erg6* expression, our findings showed that *A. fumigatus* was unable to break dormancy as indicated by conidia, showing little metabolic viability (Fig. 2A, B). Furthermore, Erg6 is not only required for germination, but also for post-germination hyphal growth. When the pTetOff-*erg6* mutant was pre-germinated under inducing conditions, germlings were unable to continue growth and support colony development when transferred to repressing culture conditions (Fig. 2C). This impaired growth induced in vitro under *erg6* repression

likely underlies the decreased mortality we noted in vivo when animals were infected with the pTetOff-*erg6* mutant and provided doxycycline (Fig. 8). As we showed that the *erg6* orthologs are essential for in vitro survival of *A. lentulus*, *A. terreus* and *A. nidulans* as well (Fig. 3), it is likely that Erg6 activity is essential for pathogenic growth across *Aspergillus* species. The importance of Erg6 to pathogenic fungal fitness appears to be conserved across many fungal pathogens, even though *erg6* orthologs are not essential in most yeast species studied to date. For example, the absence of *erg6* does not lead to growth defects in *C. glabrata*, *C. albicans*, or *K. lactis*[14–16], and causes modest or severe growth defects in *C. neoformans* and *C. lusitaniae*, respectively[17,48]. However, a *C. neoformans* Δ*erg6* mutant and mutants with reduced *erg6* expression in *C. albicans* have been reported to have significantly reduced virulence in *Galleria mellonella* infection models[17,19]. Although *erg6* null mutants in these yeast are viable, *erg6* deficiency appears to contribute to compromised phenotypes related to ergosterol-dependent functions, such as increased cell membrane permeability, reduced cell wall integrity, loss of thermotolerance and altered antifungal susceptibility profiles[14–18].

The outcomes of inhibiting specific steps in the ergosterol biosynthesis pathway are ergosterol deficiency and the accumulation of sterol intermediates. As expected, the substrate of Erg6, lanosterol, is the major accumulated intermediate when *erg6* is repressed, whereas lanosterol is a barely detectable intermediate in non-repressive conditions. Although our study indicated that the predicted Erg6 product, eburicol, remained relatively stable and ergosterol was only reduced by half in the pTetOff-*erg6* mutant under doxycycline treatment (Table 1), we attribute these outcomes to our experimental approach that partially selected for germlings displaying leaky repression in order to acquire enough biomass for study. Along with lanosterol dominating the accumulated sterol pool, cholesta-type intermediates (including cholesta-5,7,22,24-tetraenol, cholesta-5,7,24-trienol, 4,4-dimethyl cholesta-dienol and cholesta-dienol) constituted less than 4% of total sterols in pTetOff-*erg6* mutants under doxycycline treatment. These cholesta-type sterols were not detectable in non-repressive conditions (Table 1). These findings differ from the reported sterol accumulation in yeast organisms when Erg6 activity is lost. No detectable ergosterol was measured in the viable *erg6* null mutants of *S. cerevisiae*, *C. albicans*, *C. neoformans*, and *K. lactis*[14,16–18]. Instead, ergosterol biosynthesis in the absence of *erg6* causes abundant accumulation of zymosterol, cholesta-5,7,24-trienol and cholesta-5,7,22,24-tetraenol. Our findings reported here support the conclusion that these differences are likely due to variation in the preferred substrate specificities of Erg6 in different organisms. For example, *S. cerevisiae* Erg6 prefers zymosterol as a substrate, whereas lanosterol appears to be the preferred Erg6 substrate in some filamentous fungal organisms[5,12].

Probably the most explored phenotype related to *erg6* deletion in yeast is the subsequent alteration of antifungal drug susceptibility. Given that the first-line antifungal drugs, triazoles and polyenes, target ergosterol biosynthesis and ergosterol itself, respectively, we hypothesized that the defective ergosterol production resulting from *erg6* repression might affect antifungal susceptibility. Surprisingly, no significant alterations in triazole or polyene resistance profiles were observed in *A. fumigatus* in response to *erg6* repression (Fig. 7). In addition, although we noted that Erg6-GFP localization was maintained in lipid droplets upon triazole stress, *erg6* gene expression was significantly upregulated. Thus, although *erg6* is transcriptionally responsive to triazole-mediated pathway perturbation, loss of Erg6 activity does not appear to synergize with triazole therapy. The wild-type resistance profiles we observed under *erg6* repression are in stark contrast to reports from *S. cerevisiae*, *K. lactis*, *C. neoformans*, *C. albicans*, *C. auris* and *C. glabrata* in which *erg6* dysfunction is associated with increased resistance to polyenes[14–17,49,50]. This acquired polyene resistance is thought to be underpinned by ergosterol reduction or

depletion in these mutants. The resistance is then directly related to the mechanism of action of polyene drugs, which is to bind ergosterol, and extract it out of the cellular membrane to cause lethality[51]. Therefore, high MICs to polyenes are commonly seen in ergosterol-defective strains[52]. As for triazoles, the susceptibility profiles are species- and drug-dependent. Increased susceptibility has been reported for *erg6* null mutants of *S. cerevisiae*, *K. lactis* and *C. neoformans*[14,17], whereas a C. *albicans* Δ*erg6* mutant maintains wild-type susceptibility profiles[16]. *C. glabrata* and *C. auris* clinical isolates with *erg6* mutation displays increased susceptibility to triazoles[37,50], but *C. glabrata* null mutants generated in laboratory strains revealed increased tolerance[15]. Additionally, reduced resistance to triazoles has been reported in several *erg* null mutants along with ergosterol reduction or depletion[53–55]. The mechanisms involved in the alteration of triazole susceptibility caused by *erg6* mutation are complicated. The disturbed membrane fluidity and permeability caused by altered ergosterol biosynthesis, which allows azoles to penetrate abnormally, is viewed as the main potential explanation[56]. One possible explanation for the lack of changes in antifungal drug resistance profiles between wild-type and *erg6*-repressed strains in this study is that total cellular ergosterol content under the levels of *erg6* repression obtained here may still be sufficient to maintain plasma membrane functionally. It should be noted that the highest doxycycline level we used in our MIC assays was 0.25 μg/ml, which allowed observable mycelia to grow in the plates so that we could reliably measure the zone of inhibition. Therefore, this outcome could be due to the simple limitation of having to work with a doxycycline-repressible strain under conditions that still allow some level of pathway activity for viability, rather than with a complete gene deletion. A previously reported secondary mechanism of action of ergosterol biosynthesis inhibitors is in the upregulated production of reactive oxygen species within hyphae and the subsequent induction of oxidative damage[57]. Compounds that inhibit mitochondrial complex I have been shown to reduce the effectiveness of ergosterol-targeted antifungals, implying that the generation of oxidative stress is critical for the triazole antifungal effect[57]. One explanation for the lack of increased triazole susceptibility upon *erg6* repression in *A. fumigatus* may be that loss of Erg6 activity induces resistance to oxidative stress thereby abrogating this secondary mechanism. We do not believe this to be involved as previous work has shown that loss of *erg6* actually increases susceptibility to oxidative stress in *C. glabrata*[58], and repression of *erg6* expression does not result in resistance to the oxidative damage-inducing compound, menadione (Fig. S7F). Additionally, we found that *erg6* repression triggered the overexpression of two triazole resistance-associated efflux pump genes, *abcC* and *mdr1* (Fig. 7C). Therefore, increased efflux could theoretically abrogate any increased sensitivity to triazoles that may have resulted from ergosterol depletion in *erg6*-repressed strains. Further analyses of efflux pump activity changes in response to *erg6* repression are needed.

Inhibitors of sterol methyltransferase proteins have previously been the focus of intense study in the search for novel antifungal therapeutics. This is largely due to the fact that this enzymatic step does not occur in humans[59]. To date, there have been many compounds developed and tested for their sterol methyltransferase inhibitory activity against human pathogenic fungi, with the most studied being the substrate analogs that are designed as either transition state analogs, such as azasterols like 25-azalanosterol (AZAL), as well as compounds like 24(R,S),25-epiminolanosterol (EIL), or mechanism-based inhibitors, such as 26,27-dehydrozymosterol and others[59–66]. It is important to note that although these compounds have been described as having high potency and the studies describing them provide proof that Erg6 is likely a druggable enzyme target in many pathogenic fungal species, off-target toxicity has been reported with both AZAL and EIL[67–69]. While these selectivity problems have thwarted progress towards the development of Erg6 inhibitors, the recent work identifying a small molecule allosteric inhibitor with apparently minimal

cytotoxicity and patent activity against *C. albicans* Erg6 is a promising step forward[20].

In conclusion, we have validated *A. fumigatus* Erg6 as an essential protein that localizes to lipid droplets and regulates ergosterol biosynthesis. We also found that Erg6 orthologs are essential for viability in additional *Aspergillus* species in vitro and that *A. fumigatus* Erg6 is required for the accumulation of fungal burden during infection. Given the overall importance of Erg6 orthologs for growth, virulence and drug susceptibility patterns across fungal pathogens, our data support the continued development of Erg6 inhibitors as possible pan-fungal targets for novel drug development. Our findings here underpin the necessity of future work to define the ability to selectively inhibit Erg6 activity as a novel therapeutic approach for invasive aspergillosis.

## Methods

### Strains and growth conditions

All strains used in this study are summarized in Table S1. All strains were routinely cultured at 37 °C on Glucose Minimal Medium (GMM) agar plates, supplemented with 5% yeast extract, 40 μM ergosterol dissolved in ethanol or 10% fetal bovine serum as necessary[70]. Conidia were harvested from GMM plates using sterile water and stored at 4 °C.

For spot dilution assays, GMM agar plates containing doxycycline at the indicated concentrations were point-inoculated with serial dilutions of conidial suspensions from 50,000 to 50 conidia. The plates were incubated at 37 °C for 48 h. Hyphal morphology of submerged culture was assessed by inoculating $10^6$ conidia into the wells of 6-well plates containing liquid GMM at the indicated doxycycline concentrations and sterile coverslips. After 16 h at 37 °C, coverslips were washed twice with PBS and mounted for microscopy. For post-germination growth assays, $10^7$ conidia were cultured in 10 ml GMM broth for 8 h. After confirming germling formation by microscopy, ten microliters of germling suspension were inoculated onto fresh GMM plates containing doxycycline at the indicated concentrations for subculture for 48 h at 37 °C.

### Construction of mutant strains

Genetic manipulations in this study were performed using a CRISPR-Cas9 gene editing techniques described previously[25]. Briefly, for CRISPR-Cas9-mediated gene deletion, two PAM sites located upstream and downstream of the desired genes were selected and used for crRNA design. Repair templates, composed of a hygromycin resistance cassette, were amplified using primers flanked with 40 bp microhomology regions of the target locus (Table S2). For overexpression mutants, native promoters of target genes were replaced by the *hspA* promoter[25] by identifying and utilizing a single PAM site upstream of the gene coding region, as we have previously described[25]. Similarly, doxycycline-regulatable *erg6* mutants were generated via pTetOff and pTetOn promoters, as previously described[26,28]. Ribonucleoprotein (RNP) complexes were assembled in vitro using commercially available crRNA, tracrRNA, and the Cas9 enzyme as described previously[71]. Briefly, equal molar amounts of crRNA and tracrRNA were mixed in duplex buffer and boiled at 95 °C for 5 min. After cooling at room temperature for 10 min, duplex crRNA-tracrRNA was combined with Cas9 enzyme (1 μg / μl), followed by incubation for 5 min at room temperature. The transformation was performed as described previously[71]. Transformation mixtures containing 10 μl protoplasts (1–5×$10^5$ cells), 5 μl RNP (described above), repair template (900 ng), 3 μl polyethylene glycol (PEG)-CaCl₂ buffer and STC buffer (1.2 M sorbitol, 7.55 mM CaCl₂·H₂O, 10 mM Tris-HCl, pH 7.5) were incubated on ice for 50 min. Subsequently, the mixture was added to 57 μl polyethylene glycol (PEG)-CaCl₂ buffer and incubated at room temperature for 20 min. The mixture was brought to 200 μl STC buffer and plated onto a Sorbitol Minimal Medium (SMM) agar plate. After overnight room temperature incubation, transformation plates were overlaid with SMM top agar containing selective drug and incubated at 37 °C

until colonies were observed. For the generation of Erg6-GFP strain, a repair template was amplified using a GFP-expression vector that contained a linker sequence (AGATCTGGATGCGGCCGC) flanked with 40 bp microhomology regions at the 3' end of *erg6* (excluding the termination codon) to direct integration at a single downstream PAM site. All mutants were confirmed by multiple genotyping PCR reactions to ensure proper integration of the introduced repair template. All PAM sites and protospacer sequences used for crRNA design are included in Table S2. Diagnostic PCR assays to confirm mutational analyses are provided in Figure S3.

### RNA extraction and quantitative real-time PCR analysis

RNA extraction and RT-qPCR were carried out as previously described[28]. In brief, all strains were cultivated in liquid GMM supplemented with 5% yeast extract at 37 °C/250 rpm for 18 h. Mycelia were harvested, frozen in liquid nitrogen, and ground using a pestle and mortar. Total RNA was extracted using Qiagen RNeasy Mini Kit following the manufacturer's protocol. DNA contamination from RNA samples was eliminated by RNase-free Turbo DNase Kit (Invitrogen). Subsequently, cDNA was synthesized using SuperScript II system (Invitrogen), following the manufacturer's instructions. Quantitative real-time PCR was carried out using SYBR® Green Master Mix (Bio-Rad) in a CFX Connect Real-Time System (Bio-Rad).

### Antifungal susceptibility assay

The susceptibility profiles of antifungals including amphotericin B (AMB), itraconazole (ITRA), voriconazole (VOR), posaconazole (POS), and isavuconazole (IVU) were evaluated using commercial drug-embedded strips following the manufacturer's protocol and broth microdilution methodology in accordance with CLSI standard M28-A2. As for the drug strip diffusion assays, $2 \times 10^6$ conidia in 0.5 ml were spread onto GMM plates containing the indicated doxycycline concentrations. The antifungal embedded strips were applied onto the dried agar plates. After 48 h of culture, the MICs were measured by observation of the zone of clearance.

### Fluorescence microscopy

Approximately $10^6$ conidia were cultured in liquid GMM on sterile coverslips at indicated concentrations of doxycycline or antifungal drugs for 16 h at 30 °C (for ergosterol staining and viability staining) or 37 °C (for GFP analyses). For viability staining, coverslips were washed once with 0.1 M MOPS buffer (pH 3) and stained with 50 μg / ml 5,(6)-Carboxyfluorescein Diacetate (CFDA) (Invitrogen) in MOPS buffer for 1 h at 37 °C in the dark. For lipid droplet staining, coverslips were stained with 1 μg / ml BODIPY 558/568 $C_{12}$ in PBS buffer for 30 min at room temperature. For ergosterol staining, hyphae cultured on coverslips were stained with filipin (Sigma) at the final concentration of 25 μg/ml in liquid GMM for 5 min. After the above staining procedures, coverslips were washed twice with the indicated buffer and mounted for the microscope. Fluorescence microscopy was performed on a Nikon NiU microscope. CFDA staining and GFP were visualized using GFP filter settings. Lipid droplet fluorescence was captured using TRITC filter settings. Filipin staining was observed using DAPI filter settings. Images were captured by Nikon Elements software (version 4.60).

### Sterol extraction and composition analysis

Conidia were cultured in RPMI-1640 medium buffered with 0.165 M MOPS (pH 7.0) containing 0.2% w/v glucose at a final concentration of $1 \times 10^6$ cells/ml in the indicated concentrations of doxycycline for 16 h at 37 °C/250 rpm. Mycelia were harvested, and nonsaponifiable lipids were extracted as previously described[72]. Briefly, sterols were derivatized using 0.1 mL BSTFA TMCS (99:1) and 0.3 mL anhydrous pyridine and heating at 80 °C for 2 h. TMS-derivatized sterols were analyzed using GC/MS (Thermo 1300 GC coupled to a Thermo ISQ mass

spectrometer, Thermo Scientific) and identified with reference to relative retention times, mass ions, and fragmentation spectra. GC/MS data files were analyzed using Xcalibur software (Thermo Scientific). Sterol composition was calculated from peak areas, as a mean of three replicates. Data was presented as mean percentage ± SD of total sterol for each sterol.

## Murine model of invasive pulmonary aspergillosis

All animal studies were performed under the guidance of the University of Tennessee Health Science Center Laboratory Animal Care Unit and approved by the Institutional Animal Care and Use Committee under protocol 22-0373.0. Animal models of infection were performed as previously described[73]. CD-1 female mice (Charles River) weighing approximately 25 g were chemotherapeutically immune suppressed by intraperitoneal injection of 150 mg/kg of cyclophosphamide (Sigma-Aldrich) on day −3 and 75 mg / kg for subsequent injections on day +1, +4, +7, and subcutaneous injection of 40 mg / kg triamcinolone acetonide (Kenalog, Bristol-Myers Squibb) on day -1. Doxycycline was supplied at 100 mg/kg by gavage twice per day from day −3 to day +3 and once per day from day +4 to the end of study. On day 0, mice were anesthetized with 5% isoflurane and intranasally infected with a dose of $1 \times 10^6$ conidia in 20 µl saline solution. Mice were housed in sterile microisolator cages (5 animal per cage) in a university-approved animal facility on a 12 h dark/light cycle at ambient temperature and humidity. Following infection, mice monitored twice a day for 8 days. For histological study, mice were humanely euthanized by anoxia with $CO_2$ after 3 days of infection. Lungs were harvested and immediately fixed in 10% buffered formalin. Histological samples were paraffin-embedded, sectioned and stained by Grocott's Methenamine Silver.

## Statistics and reproducibility

With the exception of the survival studies involving animals, no statistical tests were utilized to pre-determine sample size. For survival studies, sample size power analyses were conducted at 80% power and an a of 0.05 to detect an anticipated incidence of mortality of 80% and 20% for the control and treated experimental arms, respectively. This analysis indicated the need for 10 animals per experimental group. As it is the minimum number of replicates required for inferential analysis, at least three biological replicates were utilized for all other experiments. No data were excluded from analyses, the experiments were not randomized, and investigators were not blinded to allocation during experiments of outcomes assessment. Statistical analyses were performed using GraphPad Prism 10.0.0 for Windows (GraphPad Software, San Diego, CA, USA). Specific tests used to determine statistical analyses are noted in each figure legend. *p* values are depicted, with a value of $p < 0.05$ considered significant.

## Reporting summary

Further information on research design is available in the Nature Portfolio Reporting Summary linked to this article.

# Data availability

All data supporting the figures included in this manuscript are provided in the Source Data file. Source data are provided with this paper.

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

## Acknowledgements

This work was supported by National Institutes of Health (NIH) / National Institute of Allergy and Infectious Diseases (NIAID) grants R01 AI158442 (JRF) and R01 AI143197 (JRF/PDR). The authors would also like to thank Nathan P. Wiederhold, PharmD at the Fungus Testing laboratory for providing the *A. lentulus* isolate used in this study.

## Author contributions

Conceptualization and planning: J.R.F, J.X.; Experimentation: J.X., W.G., A.M-V., X.G., A.V.N., H.I.T., J.E.P.; Data analysis: J.R.F, J.E.P., S.L.K., J.M.R., J.X., A.M-V., H.I.T, X.G., A.V.N., P.D.R.; Manuscript and figures preparation: J.R.F, J.X., J.E.P., S.L.K., J.M.R., A.M-V., H.I.T, W.G., X.G.

## Competing interests

The authors declare no competing interests.
