## [Peer Review File · Nature Communications]

The sterol C-24 methyltransferase encoding gene, *erg6*, is essential for viability of *Aspergillus* speciesReviewer #1 (Remarks to the Author):

This work provides a detailed and straightforward characterization of the *erg6* gene in *Aspergillus fumigatus*. It shows that Erg6 in *Aspergillus fumigatus* is essential in aspergilli, is fungal unique, and is associated with lipid droplets. Erg6 repression reduces growth and alters sterol profiles, does not alter azole or polyene susceptibility, and reduces virulence in infected mice. The manuscript is well written and generally clear.

Minor comments-

L73- please list all three enzymes absent in humans

No PCR verifications for any of the constructed strains are provided.

How were pTet off or on inserted? Ptet-off and GFP? Ptet off and haspA? With which selectable marker? Not shown in Fig S2

How was *erg6* -tet-off made in *A. nidulans*? It is resistant to hygromycin.

I cannot find Table 1 in either the main pdf or the supplementary pdf.

Reviewer #2 (Remarks to the Author):

The manuscript by Xie et al reveals that the C-24 methyltransferase (*erg6*) in *A. fumigatus* is essential for viability. The authors go on to state that Erg6 represents a potential novel drug target in this important human pathogen. Their discovery is somewhat surprising in that 1: until this study, it had been assumed that there were two functional C-24 methyltransferases in *A. fumigatus*, and because of this 2: Erg6 was dispensable for viability in fungi, and 3. Loss of C-24 methyltransferase function would alter susceptibility to the azole class of antifungals.

The vast majority of the experimental work has been performed to a very high standard and the data supports, in most part, the assertions presented by the authors. While a very compelling case has been made with respect to the essential nature of *erg6* in *A. fumigatus*, the use of *erg6* as an antifungal target is not particularly novel and more should be done to highlight this especially in the discussion (see comments below).

I do have a number issues that should be addressed:

1. The suggestion that Erg6 is a good antifungal target has not really been explored here in very much detail, other than to show that the gene is essential for viability. Little has been done in this study to define that C-24 methyltransferases are a druggable target – although there is literature to show that they can be targeted in other pathogenic fungi doi: 10.1186/1471-2180-9-74; and as cited by the authors doi.org/10.1016/j.chembiol.2023.04.010. Nether has anything been done or stated (except to say that humans lack the pathway) to show that Erg6 would be a selective target. Notably AZA and EIL, which are reported to target sterol methyltransferases are both cytotoxic...although there is some selectivity and it is likely that these compounds have off target effects. If the authors wish to state that Erg6 is a good target they need to make more efforts to confirm that this the case or at least add evidence in the discussion to highlight the significant amounts of work that have been done on this target already in other pathogenic fungi.
2. The results from the animal model of infection is not particularly convincing. The qPCR data appear highly variable and it is not obvious that adequate controls are in place to ensure that PCR inhibition is not affecting the outcome of the experiment. Often ratios between murine and fungal DNA are measured. Alternatively internal PCR controls can be used. It would be preferable to assess mortality in their murine model to ensure the apparent reduction in signal seen by qPCR results in a change in disease outcome. Finally it would be useful to show some histology of the infected murine lungs.
3. Although the data on gene essentiality is convincing, it is odd that the authors were unable to identify transformants from their attempts to create null mutants of *erg6*. Typically balanced heterokaryons are generated upon transformation of *A. fumigatus* meaning that transformants are obtained even for 'essential genes'. Can the authors comment further on this.
4. The authors state (line 138 and elsewhere) that growth was comparable between wt and modified strains, yet no quantitative data is presented to support this.

5. Line 112: % identities are given to highlight how closely related *A. fumigatus* proteins (*erg6* and *smt1*) are to those of *S. cerevisiae*. The authors should also include details of the % coverage of the protein.
6. The data showing that the *erg6* null is not differentially susceptible to the azoles is not very convincing. The data appear to be highly variable. Of specific note the halo around VOR DOX 0.25 ug/ml seems quite large yet the stated MIC hardly changes. Can the authors explain this. I note that MICs have been conducted in microtiter plates – perhaps evidence showing AUC could be shown to show if there are modest changes to susceptibility at sub-MIC values.
7. Could the authors comment further as to why susceptibility to the azoles is unchanged. Could this be linked to the apparent role of heightened oxidative stress in the mechanism of action of the azoles? See doi: 10.1128/AAC.00978-17
8. Fig4 legend -typo – reads Fig. 3 *Erg6*
9. I am aware that it probably wasn't the intention of the authors – but line 65 seems to suggest that polyenes target ergosterol biosynthesis. This sentence should be modified.
10. Line 89 -existing rather than exiting.

Reviewer #3 (Remarks to the Author):

Dear authors,

The article intitled (The sterol C-24 methyltransferase encoding gene, *erg6*, is essential for viability of *Aspergillus* species provides new insights on C24-methyltransferases encoding gene, *erg6*, in *Aspergillus* species. Until now there were many informations on C24-methyltransferases among yeasts but few was done in *Aspergillus* species.

It is common for the genome of filamentous fungi to encode multiple paralogous genes in the ergosterol synthesis pathway. In this study *erg6* and *smt1* were characterized as putative sterol C24-methyltransferases encoding genes. *smt1* appears to play a minimal role in ergosterol biosynthesis and/or *erg6* or *erg6* activity is able to compensate for loss of *smt1*. On the contrary, *erg6* deficiency is lethal for *Aspergillus fumigatus* and *erg6* activity is essential for pathogenic growth. The same effect is observed for its orthologs in *Aspergillus* species such as *A. lentulus*, *A. terreus* and *A. nidulans*.

Erg6 is an essential protein for growth and viability at every stage of growth (germination and post germination hyphal growth) in vitro and is also required for fungal infection establishment in a murine model. When *erg6* expression is repressed a loss of ergosterol is observed in the membrane and an accumulation of lanosterol is observed, due to its role in an early step of ergosterol biosynthesis in *Aspergillus* sp, i.e. converting lanosterol into eburicol. A loss of *Erg6* at the protein level has been correlated to the repression of *erg6* gene expression, but its localization stays in the lipid droplets. Unfortunately, downregulation of *erg6* does not drive to modification of triazole and polyene susceptibility profiles contrary to what is observed among other fungi. An overexpression of *abcC* and *mdr1* genes encoding for efflux pumps has been observed under *erg6* repression conditions and that can counterbalance the accumulation of antifungals on fungal cells. All these results support the idea that *Erg6* is a promising fungal target for novel drug development.

This article is really interesting and is worth publishing but some corrections and improvements are mandatory particularly concerning the sterol analysis part. There is a confusion between total sterols and 24-methylated sterols. That is not the same thing and the difference has to be considered by the authors, as proposed below in the major revision part. Then some details could be more explained or discussed or some minor mistakes have to be corrected, this is detailed in the minor concerns part

Major reviews:

Concerning the sterol part the confusion between total sterols and C24-methylated sterols has to be corrected :

L227 : harvested is a wrong term! There is a chemical reaction with trimethylsilane it's not just harvested. You should change for trimethylsilylation or derivatization with trimethylsilane

L230 : It is the same mistake as in the legend of the table 1. Ergosterol is accounting for nearly

90% of total sterols and not only 24-methylated sterols. The sentence is wrong as written in the text.

L235-236 : Here again, the amount of lanosterol is nearly 40 % of total sterols and not only the 24-methylated. The sentence is wrong as written in the text.

L238 : Here again, the amount of cholesta type intermediates accounted for less than 4 % of total sterols and not only the 24-methylated. The sentence is wrong as written in the text.

L387-388 : in fact the reason of the differing sterol profiles between *A.fumigatus* and yeast under *erg6* deficiency doesn't seem unclear. As a matter of fact, the explanation is in the two next sentences and in your work in this paper. Looking to the sterol biosynthesis pathway in different organisms and the preferred substrate specificities of Erg6, you can explain the differences. You have confirmed in this paper that *Aspergillus fumigatus erg 6* has a conserved role in ergosterol biosynthesis at the lanosterol-eburicol conversion step (l 240-242). You have to compare with pathways described in *Candida* for exemple where Erg6 act at the zymosterol-fecosterol conversion step.

L518 : add "of total sterol" to the last sentence. "Data were presented as mean percentage (\pm SD) of total sterol for each sterol

Table 1 : The title should be modified, here are studied all the sterols not only the 24-methylated, it's indicated below the table. The title should become: Changes in total sterol composition in response to *erg6* repression

Can you explain how are classified the sterols in the table. If it is thanks to their relative retention time (as often in this kind of studies), it would be interesting to write it.

One can ask if it is really relevant to give percentage of 0.7 \pm 0.5 or 0.5 \pm 0.5%. As it is written, the sterols that are relevant have a % > 2%. Maybe for the others it's not necessary to give the percentage, only to say that it is less than 2% and that they are detected. The amount itself doesn't give so much information.

In the material and method part, the reference chosen for the sterol extraction and composition is not the best one (ref 61), because it refers to another genus (*Candida glabrata*) and it is not the same way to make the culture and to obtain the sterols

Minor reviews:

Figure 2, Part B: why is the microscope magnification not the same for the pTetOn-*erg6* and the pTetOff-*erg6*. Is it normal? Is there an explanation?

Figure 4: There is a problem with the legend. You have to remove Fig. 3 Erg6

Fig 5: why is it written in the materials and methods part that the temperatures of incubation are 30°C or 37°C whereas in the legend of this figure it is only 30°C?

Fig S5 and fig 7: Is it usual to find such a difference in the MIC values obtained by the strip diffusion assays and the CLSI standard M38-A2, particularly for itraconazole, posaconazole and amphotericin B ? Fig 7, can you precise the temperature of incubation?

Fig S2: box A, Δ erg6 should be written on one line and not split between Δ erg and 6

Fig S4 : Do you have an explanation for the difference in the size of the spots obtained with the pTetOn-*erg6* compared to the parental one, even with a high dose of doxycyclin.

Fig S6: There is a problem with the legend. You have to remove Fig. S5

Fig S6A and S6B: Why is this figure appearing only in the discussion part end not in the results one? It could be interesting to analyse the results before the discussion, as for all the others figures

Improvement

It should be interesting to analyse the sterols profile in *smt1* mutants (Δ -*smt1* or OE *smt1*) to verify if that enzyme contributed to C24-methyltransferase activity as asserted in the discussion line 340-347.

RESPONSE TO REVIEWER COMMENTS

Reviewer #1

Comment: L73- please list all three enzymes absent in humans

Response: We have edited the manuscript to list the pertinent sterol biosynthesis enzymes absent in humans. LINE 73.

Comment: No PCR verifications for any of the constructed strains are provided.

Response: We have now included all diagnostic PCR assays for the constructed strains in a new Figure S3 and have referenced this figure in the Materials and Methods section describing mutant strain construction. LINES 538-539, 702-703.

Comment: How were pTet off or on inserted? Ptet-off and GFP? Ptet off and haspA? With which selectable marker? Not shown in Fig S2

Response: We apologize for the overly simplistic presentation of genetic manipulations in our initial submission and have updated Figure S2 to include this important information.

Comment: How was *erg6* –tet-off made in *A. nidulans*? It is resistant to hygromycin.

Response: Again, we apologize for this oversight. Genetic manipulations involving the TetOff promoter insertion in *A. nidulans* were carried out using a phleomycin resistance cassette. This is now noted in Figure S2.

Comment: I cannot find Table 1 in either the main pdf or the supplementary pdf.

Response: Please accept our apologies for this omission. Table 1 was submitted for review upon request by the editor.

Reviewer #2

Comment: The suggestion that Erg6 is a good antifungal target has not really been explored here in very much detail, other than to show that the gene is essential for viability. Little has been done in this study to define that C-24 methyltransferases are a druggable target – although there is literature to show that they can be targeted in other pathogenic fungi doi: 10.1186/1471-2180-9-74; and as cited by the authors doi.org/10.1016/j.chembiol.2023.04.010. Nether has anything been done or stated (except to say that humans lack the pathway) to show that Erg6 would be a selective target. Notably AZA and EIL, which are reported to target sterol methyltransferases are both cytotoxic...although there is some selectivity, and it is likely that these compounds have off target effects. If the authors wish to state that Erg6 is a good target they need to make more efforts to confirm that this the case or at least add evidence in the discussion to highlight the significant amounts of work that have been done on this target already in other pathogenic fungi.

Response: We thank the reviewer for these very important points. As our study is only first step in validating Erg6 as potential antifungal target in *A. fumigatus*, we agree that we have not yet provided biochemical evidence that Erg6 can be selectively targeted for therapy. To address this, we have softened the language throughout referring to *A. fumigatus* Erg6 as a “good drug target” and have included a new section in the Discussion to include the reviewer’s points. LINES 473-486.

Comment: The results from the animal model of infection is not particularly convincing. The qPCR data appear highly variable and it is not obvious that adequate controls are in place to ensure that PCR inhibition is not affecting the outcome of the experiment. Often ratios between murine and fungal DNA are measured. Alternatively internal PCR controls can be used. It would be preferable to assess mortality in their murine model to ensure the apparent reduction in signal seen by qPCR results in a change in disease outcome. Finally it would be useful to show some histology of the infected murine lungs.

Response: We agree with the reviewer that assessing mortality is a better indicator or the importance of *erg6* to disease initiation and progression. We also agree that histology is an incredibly useful readout of infection in this model. Therefore, we have completed new survival analyses using our pTetOff-*erg6* mutant (with parental controls) in the presence and absence of doxycycline administration. The survival results, coupled with the histology, show that *erg6* is required for infection. As doxycycline administration is not always uniform across mice in studies such as these, we have further complemented this work by inclusion of the pTetOn-*erg6* mutant with no doxycycline administration. Our results provide strong evidence that a strain lacking *erg6* expression is unable to initiate infection. These new data now replace the original fungal burden study in Figure 8. We have updated the Results (LINES 315-352), Figure Legend (LINES 666-675), and Materials and Methods (LINES 583-595) sections.

Comment: Although the data on gene essentiality is convincing, it is odd that the authors were unable to identify transformants from their attempts to create null mutants of *erg6*. Typically, balanced heterokaryons are generated upon transformation of *A. fumigatus* meaning that transformants are obtained even for ‘essential genes’. Can the authors comment further on this.

Response: The reviewer is correct that balanced heterokaryons are sometimes isolated when targeting essential genes for deletion using traditional gene targeting methods. Using

the CRISPR/Cas9-mediated approaches described in this paper, we have found that we rarely recover heterokaryons and believe this to be a product of the highly efficient targeting and recombination that this system provides. In conjunction with the lack of colonies from gene deletion attempts, our pTetOff and pTetOn data clearly support the essential nature of Erg6 to *A. fumigatus* viability.

Comment: The authors state (line 138 and elsewhere) that growth was comparable between wt and modified strains, yet no quantitative data is presented to support this.

Response: We have now included quantitative data in a new Figure S4 (LINES 705-707). For the different *Aspergillus* species we tested, we now show that *A. nidulans* and *A. lentulus* pTetOff-*erg6* strains grow similarly to the parental controls in the absence of doxycycline. *A. terreus*, however, displayed a significant colony diameter reduction after 48 hrs of culture under the same conditions. This new data is referenced at LINES 196-200.

Comment: Line 112: % identities are given to highlight how closely related *A. fumigatus* proteins (*erg6* and *smt1*) are to those of *S. cerevisiae*. The authors should also include details of the % coverage of the protein.

Response: We apologize for this oversight and have now included the % coverage information. To make these comparisons as relevant as possible for *Smt1*, we have also clarified the % identity and coverage specifically within the conserved functional domains of each protein. This new information is included now in Fig S1B (LINES 684-688) and in the text at LINES 115-125.

Comment: The data showing that the *erg6* null is not differentially susceptible to the azoles is not very convincing. The data appear to be highly variable. Of specific note the halo around VOR DOX 0.25 ug/ml seems quite large yet the stated MIC hardly changes. Can the authors explain this. I note that MICs have been conducted in microtiter plates – perhaps evidence showing AUC could be shown to show if there are modest changes to susceptibility at sub-MIC values.

Response: We thank the reviewer for this comment and hope we can bring clarity with the following. First, for defining “susceptibility” we are using the strict measurement of shifts in MIC (reported as 100% growth inhibition) under the conditions tested. The difference in halo size at higher concentrations of doxycycline is due to growth inhibition that occurs during *erg6* repression in the pTetOff-*erg6* strain. The MIC of a drug strip assay, however, is read at the highest point at which visible fungal growth intersects the strip. The halo size does not necessarily impact this endpoint. This is why MIC values remained mostly unchanged in the drug strip assays while the halo size differs to some extent. However, because there was halo size variability, we also complemented those findings using broth microdilution MIC assays performed by published CLSI-methodology. This is considered the “gold standard” for MIC analyses. The results of both show that limiting *erg6* expression, at least to the point that still permits enough fungal growth to perform the assays, does not significantly impact MIC to triazoles or amphotericin B. Variability between strip diffusion and BMD-based assays has been described previously for filamentous fungi (PMID: 26202113; PMID: 28994001). The goal of these studies was to test if loss of *erg6* would cause the same phenotypes in *A. fumigatus* that have been reported in yeast organisms. For example, deletion of *erg6* in *Cryptococcus neoformans* results in a 64 to 128-fold decrease in MIC to triazoles and a 4-fold increase in MIC to amphotericin B. As can be seen, we do not detect this level of MIC shift in our *erg6*-repressed strain. The major limitation here is that we are not able to fully

delete *erg6* from *Aspergillus* species (as can be done in many yeast), so there remains a possibility that significant drug susceptibility changes require complete loss of Erg6 activity. We have noted this limitation in the Discussion. LINES 456-458.

Comment: Could the authors comment further as to why susceptibility to the azoles is unchanged. Could this be linked to the apparent role of heightened oxidative stress in the mechanism of action of the azoles? See doi: 10.1128/AAC.00978-17.

Response: Lines 417-455 are largely text of the Discussion section from the initial submission that covered possible reasons for the lack of increased susceptibility. One of the major limitations with this part of our study is that we have to work with a strain that expresses enough *erg6* to support viability and, therefore, may not fully shutdown this point of the ergosterol biosynthesis pathway. In most yeast organisms, *erg6* is non-essential. This allows complete gene deletion and subsequently results in a strain that has no detectable ergosterol and significant accumulation of many sterol intermediates. The possibility exists that we cannot re-reproduce the same level of pathway dysfunction in *A. fumigatus* and still have a viable strain for analyses. Though this was mentioned in the previous submission, we added text to clarify this limitation. LINES 456-458. If the secondary mechanism of action involving triazole-induced oxidative stress were at play here, we would expect that downregulation of *erg6* would at least partially improve resistance to oxidative stress (therefore, offsetting oxidative stress induced by pathway inhibitors). We believe this to be unlikely, given previous work showing that deletion of *erg6* has actually been associated with decreased ability to survive oxidative stress in *Candida* (PMID: 37233290). To further confirm this, we now provide new evidence in Supplementary Figure 7F (LINES 727-729), showing that down regulation of *erg6* does not generate oxidative stress resistance in *A. fumigatus*. We have added text to the Discussion regarding this point. LINES 458-468.

Comment: Fig4 legend -typo – reads Fig. 3 Erg6

Response: Corrected.

Comment: I am aware that it probably wasn't the intention of the authors – but line 65 seems to suggest that polyenes target ergosterol biosynthesis. This sentence should be modified.

Response: Thank you for the opportunity to clarify. We have corrected this sentence to clear up any ambiguity. LINES 65-66.

Comment: Line 89 -existing rather than exiting.

Response: Corrected.

Reviewer #3:

Comment: Concerning the sterol part the confusion between total sterols and C24-methylated sterols has to be corrected.

Response: We apologize for the consistent oversight on our part. Where appropriate, we have corrected the term “C24-methylated sterols” to just “total sterols”. These changes address multiple critiques raised by the reviewer below.

Comment: L227 : harvested is a wrong term! There is a chemical reaction with trimethylsilane it's not just harvested. You should change for trimethylsilylation or derivatization with trimethylsilane

Response: We thank the reviewer for catching this error and giving us the opportunity to be specific. We have corrected the sentence to read that total sterols were “...derivatized to trimethylsilyl ethers and analyzed...”(LINE 241).

Comment: L230 : It is the same mistake as in the legend of the table 1. Ergosterol is accounting for nearly 90% of total sterols and not only 24-methylated sterols. The sentence is wrong as written in the text.

Response: Corrected. Please see response to the first comment made by Reviewer 3. LINE 244.

Comment: L235-236 : Here again, the amount of lanosterol is nearly 40 % of total sterols and not only the 24-methylated. The sentence is wrong as written in the text.

Response: Corrected. Please see response to the first comment made by Reviewer 3. LINE 249.

Comment: L238 : Here again, the amount of cholesta type intermediates accounted for less than 4 % of total sterols and not only the 24-methylated. The sentence is wrong as written in the text.

Response: Corrected. Please see response to the first comment made by Reviewer 3. LINE 252.

Comment: L387-388 : in fact the reason of the differing sterol profiles between *A. fumigatus* and yeast under *erg6* deficiency doesn't seem unclear. As a matter of fact, the explanation is in the two next sentences and in your work in this paper. Looking to the sterol biosynthesis pathway in different organisms and the preferred substrate specificities of Erg6, you can explain the differences. You have confirmed in this paper that *Aspergillus fumigatus erg6* has a conserved role in ergosterol biosynthesis at the lanosterol-eburicol conversion step (l 240-242). You have to compare with pathways described in *Candida* for example where Erg6 act at the zymosterol-fecosterol conversion step.

Response: We thank the reviewer for noting that our findings reported herein (as well as those we have referenced in the literature) directly support the conclusion that these differences are largely due to varying substrate specificities between yeast and *Aspergillus Erg6* enzymes. We have corrected the text to state this. LINES 421-424.

Comment: L518 : add “of total sterol” to the last sentence. “Data were presented as mean percentage (\pm SD) of total sterol for each sterol

Response: We have corrected this sentence as suggested. LINE 581.

Comment: Table 1: The title should be modified, here are studied all the sterols not only the 24-methylated, it's indicated below the table. The title should become: Changes in total sterol composition in response to erg6 repression

Response: We, again, thank the reviewer for catching this error and have corrected the Table 1 title. LINE 598.

Comment: Can you explain how are classified the sterols in the table. If it is thanks to their relative retention time (as often in this kind of studies), it would be interesting to write it.

Response: This information is included in the Materials and Methods (LINES 576-579) where the sterols are described as being classified “...with reference to relative retention times, mass ions, and fragmentation spectra.”

Comment: One can ask if it is really relevant to give percentage of 0.7+/-0.5 or 0.5+/-0.5%. As it is written, the sterols that are relevant have a % > 2%. Maybe for the others it's not necessary to give the percentage, only to say that it is less than 2% and that they are detected. The amount itself doesn't give so much information.

Response: Although we think the reviewer has a fair point, we have chosen to leave these sterols in the provided table for the sake of being complete in our analysis.

Comment: In the material and method part, the reference chosen for the sterol extraction and composition is not the best one (ref 61), because it refers to another genus (*Candida glabrata*) and it is not the same way to make the culture and to obtain the sterols.

Response: The culture conditions under which the *Aspergillus* strains were grown is clearly described in this section. Although the provided reference is from studies using *Candida*, the method for sterol extraction was performed in exactly the same manner.

Comment: Figure 2, Part B: why is the microscope magnification not the same for the pTetOn-erg6 and the pTetOff-erg6. Is it normal? Is there an explanation?

Response: We utilized a wider field of view for the pTetOff mutant so the reader can appreciate the changes in viability between doxy and no doxy conditions (quantitated in Figure 2A). For the pTetOn mutant, we used high magnification to display the unpolarized conidia more clearly in the absence of doxycycline.

Comment: Figure 4: There is a problem with the legend. You have to remove Fig. 3 Erg6

Response: Corrected.

Comment: Fig 5: why is it written in the materials and methods part that the temperatures of incubation are 30°C or 37°C whereas in the legend of this figure it is only 30°C?

Response: We have clarified in the Materials and Methods at LINES 559-560. Ergosterol staining and viability staining were performed at 30°C, whereas GFP analyses were performed at 37°C.

Comment: Fig S5 and fig 7: Is it usual to find such a difference in the MIC values obtained by the strip diffusion assays and the CLSI standard M38-A2, particularly for itraconazole, posaconazole and amphotericin B? Fig 7, can you precise the temperature of incubation?

Response: With regards to the variability in strip diffusion assays, please see the response to Reviewer #2 which we hope adequately answers the reviewer's concerns. We have now included the temperature of incubation for Fig 7 in the figure legend.

Comment: Fig S2: box A, deltaerg6 should be written on one line and not split between deltaerg and 6

Response: Corrected.

Comment: Fig S4 : Do you have an explanation for the difference in the size of the spots obtained with the pTetOn-erg6 compared to the parental one, even with a high dose of doxycyclin.

Response: This is likely due to the inability of the pTetOn promoter system to achieve wild type *erg6* gene expression levels, even under high exogenous doxycycline. Each of the promoter systems display variations in the ability to induce/repress alleles, which is evident when comparing the pTetOn-erg6 strain (with doxy) to the pTetOff-erg6 strain (without doxy).

Comment: Fig S6: There is a problem with the legend. You have to remove Fig. S5

Response: Corrected.

Comment: Fig S6A and S6B: Why is this figure appearing only in the discussion part and not in the results one? It could be interesting to analyse the results before the discussion, as for all the others figures.

Response: We have now moved the description of this dataset to the Results section at LINES 291-297.

Comment: Improvement - It should be interesting to analyse the sterols profile in *smt1* mutants (delta-*smt1* or OE *smt1*) to verify if that enzyme contributed to C24-methyltransferase activity as asserted in the discussion line 340-347.

Response: We thank the reviewer for suggestion for improvement. We have edited the language slightly at LINES 379-380 to more clearly state that *Smt1* "...does not contribute to C24-methyltransferase activity to the extent that it can compensate for loss of *Erg6*" and have acknowledged the need for additional sterol profiling to fully delineate the cellular role(s) for *Smt1*. We see this as something to pursue in future follow up studies on fungal *Smt1* orthologs, especially given the predicted lack of conservation of the C-terminal sterol methyltransferase domain in the *A. fumigatus* we have reported in Figure S1B.

Reviewer #1 (Remarks to the Author):

The authors have addressed all my comments appropriately and revised the MS accordingly

Reviewer #2 (Remarks to the Author):

The modifications to the manuscript have resulted in significant improvements both to the clarity and robustness of this excellent study.

I have no further comments.

Reviewer #3 (Remarks to the Author):

I wanted to thank the authors for their answers to my questions and their efforts to amend all the points I had raised.

The manuscript is now ready for publication, from my point of view.